# The Association of Cognitive Impairment and Depression with Malnutrition among Vulnerable, Community-Dwelling Older Adults: A Large Cross-Sectional Study

**DOI:** 10.3390/geriatrics9050122

**Published:** 2024-09-19

**Authors:** George Karam, Nada Abbas, Lea El Korh, Alexander Abi Saad, Lara Nasreddine, Krystel Ouaijan

**Affiliations:** 1Department of Psychiatry and Clinical Psychology, St. George Hospital University Medical Center, Beirut 11002807, Lebanon; gekaram@stgeorgehospital.org (G.K.);; 2Institute for Development, Research, Advocacy and Applied Care (IDRAAC), Beirut 11002110, Lebanon; 3Faculty of Medicine, Saint George University of Beirut, Beirut 11002807, Lebanon; 4Department of Nutrition and Food Sciences, Faculty of Agriculture and Food Sciences, American University of Beirut, Beirut 11072020, Lebanon

**Keywords:** nutrition screening, malnutrition, cognitive impairment, dementia, depression, older adults, community-dwelling

## Abstract

(1) Background: Mental health issues in older adults, particularly cognitive impairment and depression, can affect nutritional status. This study investigates the prevalence of malnutrition among community-dwelling older adults at risk of social exclusion and dependency in Lebanon and its association with cognitive impairment and depression. (2) Methods: This cross-sectional study used secondary data from the TEC-MED project, involving 1410 older adults aged 60 and above in Beirut. Nutritional status was assessed with the Mini Nutritional Assessment _Short Form (MNA_SF), cognitive impairment with the AD8 Dementia Screening Interview, and depression with the Geriatric Depression Scale (GDS-15). (3) Results: 87.2% of participants were at risk of malnutrition, and 2.5% were malnourished. Cognitive impairment was present in 82.2% of the sample and 45% experienced moderate to severe depression. Malnourished individuals had significantly higher rates of cognitive impairment (96.7% vs. 57.8%) and depression (85.7% vs. 23.2%). Significant associations were found between the risk of malnutrition, decreased food intake, cognitive impairment, and depression; however, no significant association was found with BMI. Logistic regression analysis indicated that older age, cognitive impairment, and depression were significant predictors of malnutrition, while having a caregiver was protective. (4) Conclusions: The high prevalence of risk of malnutrition among vulnerable older adults with cognitive impairment and depression underscores the need for policies integrating nutritional screening into routine health check-ups for older adults.

## 1. Introduction

Dementia, defined as a progressive neurodegenerative condition that affects cognitive function, is a common mental health disorder that affects a large proportion of the aging population [1,2]. The estimated global prevalence of dementia in older adults aged 60 years and more is between 5 and 7% with a higher burden in low- and middle-income countries [1]. This high global prevalence, equivalent to 55 million people in 2020, is predicted to double every 20 years to reach 139 million by 2050 [3]. The increase in the aging population worldwide coupled with the significant impact of cognitive decline and dementia on social and health conditions, makes dementia a major public health priority [4].

A major consequence of cognitive decline and dementia is their significant impact on the nutritional status of older adults [5]. Factors such as decreased ability in meal planning, taste alterations, and swallowing difficulties result in decreased food intake, weight loss, and consequently malnutrition [6,7]. In turn, malnutrition is associated with severe complications such as reduced immunity and loss of functional capacity, which may result in increased mortality [8,9]. Despite these severe consequences, nutritional status is not systematically screened in older adults with dementia [10].

Based on available studies, older adults with cognitive impairment and dementia living in long-term care facilities are at a higher risk of malnutrition compared to those without dementia, with prevalence estimates ranging from 6.8% to 75% [2]. This wide discrepancy in prevalence can be attributed to the use of different diagnostic tools for assessing malnutrition, as well as variations in socioeconomic conditions and geographic locations across study populations [2]. Depression has been associated with malnutrition in older adults living in long-term care institutions [11,12]. Additionally, studies have shown a link between cognitive decline and the severity of nutritional risk in institutionalized older adults, particularly in those with depression [11,13]. However, very few studies have been conducted on older adults living in their own homes in the community and who suffer from cognitive impairment [14,15,16]. Among these, a recent study conducted in China has suggested a positive relationship between malnutrition and cognitive decline among community-dwelling older adults aged 80 years and above [15]. Addressing this gap in research is vital, as early screening and interventions can potentially mitigate the adverse effects of malnutrition in older adults with cognitive impairment [6,17].

In the countries of the Middle East, including Lebanon where our study was conducted, demographic transition is leading to an aging population, similar to in other parts of the world [18]. This aging process is supported by strong social norms that encourage families to make living arrangements to keep their older adults at home, prioritizing cultural habits of having older adults living with their family or in their own homes [18,19]. These efforts lead to arrangements like multi-generational households, family caregivers, or home-based nurses, contributing to a higher proportion of older adults living in the community rather than in long-term care facilities [19].

It is in this context our study was conducted, with the aims of (1) examining the prevalence of both malnutrition and dementia among older adults living in the community in Lebanon; (2) determining the basic epidemiological indicators of dementia, depression, and malnutrition, and investigating their association to guide early detection and timely treatment; and (3) identifying the most common indicators of malnutrition among Lebanese older adults. By achieving these objectives, our study can lay the foundation for the development of policies and interventions that target the early screening of malnutrition in older adults with dementia, ultimately improving their quality of life.

## 2. Materials and Methods

### 2.1. Study Design and Population

This observational, community-based cross-sectional study used secondary data from Lebanon’s cohort within the Transcultural Socio-Ethical-Care Model for Dependent Populations in the Mediterranean Basin (TEC-MED), which is part of the 2019–2023 CrossBorder Cooperation (CBC) initiative, funded by the European Neighborhood Instrument (ENI) [20]. The project aims to create a transcultural care approach for dependent elderly people in six Mediterranean countries (Spain, Greece, Italy, Tunisia, Egypt, and Lebanon) and includes an initial assessment using an integrated platform, followed by social care interventions.

In Lebanon, participants were recruited through convenience sampling from primary care centers, religious institutions, and referrals in the capital Beirut. Eligible participants were older adults (60 years and above) living at home and classified as dependent or at risk of dependency, based on the Inventario del NIvel de Cuidados mediante Indicadores de Resultados de Enfermería (INICIARE), and/or at risk of poverty and social exclusion based on the At Risk of Poverty or Social Exclusion (AROPE) criteria. The INICIARE tool short form was used to measure dependency levels by assessing functional capabilities and support needs, including feeding, continence, and health behavior [21]. AROPE is an EU policy tool used to assess individuals or households at risk of poverty, severe material deprivation, or living in households with low socioeconomic status [22]. Exclusion criteria included severe or total dependency according to INICIARE and/or AROPE, needing palliative or end-of-life care, uncorrected severe hearing loss, uncontrolled severe mental illness, and those taking major tranquilizers or other drugs affecting cognitive status. A total of 4259 participants met the criteria and were enrolled in the TEC-MED project in Lebanon. However, our study specifically included only the 1410 participants who completed the nutritional assessment.

### 2.2. Data Collection

Data collection occurred between April 2022 and February 2023 through home visits by trained social workers, who conducted an initial assessment. The social workers underwent comprehensive training by physicians and dietitians on conducting nutritional assessments and using them to assess mental health, with a focus on cognitive impairment and depression. When applicable, caregivers providing care to these older persons at home were also involved in the data collection. Data on the patient’s characteristics, including age, gender, comorbidities, and the presence of a caregiver were also recorded [21].

### 2.3. Nutritional Assessment

Nutritional status was assessed using the Mini Nutritional Assessment_Short Form (MNA_SF) scale [23]. The MNA_SF is a validated short version of the 18-item MNA malnutrition assessment tool questionnaire. It consists of six questions on decreased food consumption, weight loss, mobility, the presence of acute disease, neuropsychological problems (including dementia and depression), and anthropometry measured by BMI. Participants were classified as normal or well-nourished if their score was ≥12 points, at nutritional risk if their score was 8–11 points, and malnourished if their score was ≤7 points.

### 2.4. Assessment of Mental Health and Cognitive Function

Cognitive impairment and dementia symptoms were measured using the validated Ascertain Dementia (AD8) screening tool [24]. It includes 8 items that ask about various related signs such as memory deficits, decreased interest in hobbies, and trouble learning, among others. Possible answers are “Yes, some change”, “no change”, and “don’t know”. Scores of 0 or 1 are considered indicative of normal cognition, while higher scores (2–8) are indicative of some cognitive decline.

Current depressive symptoms were measured using the Geriatric Depression Scale (GDS-15) [25,26]. The scale comprises 15 yes/no items that ask about various related symptoms such as life satisfaction, mood, boredom, and emptiness, among others. All items are scored equally, 1 for yes and 0 for no, with a total sum between 0–15. Scores ≤4 were considered normal, 5–8 indicative of mild depressive symptoms, 9–11 indicative of moderate depressive symptoms, and ≥12 indicative of severe depressive symptoms [25,26].

### 2.5. Statistical Analysis

Since this study is a secondary analysis of a large dataset (TEC-MED), a post-hoc power analysis was conducted to compute the power level (1-β) using the given sample size and a significance level (α) of 0.001. The analysis was performed using the G*Power statistical tool, and the resulting power was greater than 99% [27]. All statistical analyses were conducted using Stata Version 15 for Windows. A *p*-value < 0.05 was considered significant. Continuous variables were presented as mean ± SD, and categorical variables as frequency and percentage.

When analyzing the association between malnutrition and dementia, the scale was modified by removing the dementia and depression item (scored 0–2). Based on Sanders et al., the score cutoffs were adjusted: 10–12 for normal nutrition, 7–9 for risk of malnutrition, and 6 or below for malnutrition [16]. This modification was applied only for correlation and regression models to avoid confounding elements. Bivariate associations were tested using the chi-square test. Simple and multiple logistic regression analyses were used to identify determinants of MNA. Variables with a significant *p*-value in the bivariate analysis were included in the multiple regression model.

### 2.6. Ethical Approval

This study was conducted according to the guidelines laid down in the Declaration of Helsinki and all procedures involving human subjects/patients were approved by the Institutional Review Board (IRB) of the University of Balamand, IRB-REC/O1051-2112321. Written informed consent was obtained from all participants and their caregivers.

## 3. Results

### 3.1. Sample Description

From the initial cohort of 4259 participants, 1410 participants had complete nutritional assessment data and were included in the study. The average age of the participants was 71.6 ± 7.9 years and 64% of the study group were females. The most prevalent comorbid condition observed was diabetes, affecting 15.1% of the population. Additionally, 4.3% of the participants had multiple comorbidities, defined as having more than one medical condition according to the Charlson Comorbidity Index [28]. Baseline characteristics are presented in Table 1.

### 3.2. Prevalence of Malnutrition, Cognitive Impairment, and Depression

Table 2 presents the Mini Nutritional Assessment_Short Form (MNA_SF) scale items and total scores. A significant majority reported a moderate decrease in food intake (90.7%). Regarding recent weight changes, 88.4% were uncertain about weight loss, while 9.9% reported no recent weight loss and a minority (1.7%) reported “any weight loss” (≥1 kg). Mobility assessments indicated that 1.6% of participants were bed or chair bound, 9.8% were able to get out of bed or their chair but were unable to leave their home, and 88.7% reported being able to leave their home. Psychological stress or acute disease in the past three months was reported by 3.0% of the cohort. Analysis of BMI distribution showed an average BMI of 24.2 kg/m^2^ (±3.79), with 54.6% of participants having a BMI of 23 or greater. The MNA_SF total scores revealed that 87.2% of participants were at risk of malnutrition, while 10.3% had a normal nutritional status. Malnutrition was observed in 2.5% of individuals.

The AD8 Dementia Test indicated an average score of 4.50 (±2.72). Of the study sample, 17.8% had normal cognition (AD8 score 0–1), while 82.2% showed signs of cognitive impairment (AD8 score 2–8). The Geriatric Depression Scale was completed by only 729 participants. It revealed an average score of 8.30 (±2.15). A small proportion (5.9%) fell within the normal or asymptomatic range (GDS score 0–4), while 48.1% exhibited mild depression (GDS score 5–8), 40.7% showed moderate depression (GDS score 9–11), and 5.2% had severe depression (GDS score 12–15). The prevalence rates of malnutrition, dementia, and depression are illustrated in Figure 1. There was no significant association between MNA_SF and sex (*p* = 0.748). Similarly, no significant differences across sexes were observed for dementia (*p* = 0.221) or depression (*p* = 0.062).

### 3.3. Rates of Malnutrition among Older Adults with Cognitive Impairment and Depression

Among older adults with normal cognition, 42.2% had normal nutritional status while 57.8% were either at risk of malnutrition (52.6%) or malnourished (5.2%). In contrast, among individuals with cognitive impairment, only 3.4% had normal nutritional status, while the majority (96.7%) were either at risk of malnutrition (94.7%) or malnourished (2%). The difference between the two groups was statistically significant (*p*-value < 0.001) (Figure 2).

Regarding depression, among those with a normal score, 76.7% had normal nutritional status, 20.9% were at risk of malnutrition, and 2.3% were malnourished. Conversely, among older adults with depression, 14.3% had normal nutritional status while 85.7% were either at risk of malnutrition (81.8%) or malnourished (3.9%). The difference between the two groups was statistically significant (*p*-value < 0.001) (Figure 2).

### 3.4. Association of Malnutrition and Modified MNA_SF and MNA_SF Criteria with Cognitive Impairment and Depression

Table 3 and Table 4 present the distribution of participants across MNA_SF criteria, comparing those with normal cognition (AD8 score 0–1) to those with cognitive impairment (AD8 score 2–8), and those with no depression (GDS score 0–4) to those with mild, moderate, or severe depression (GDS score 5–15). The modified MNA scale, omitting the dementia and depression item, was used when analyzing associations with dementia and the score cutoffs were adjusted to 10–12 for normal nutrition, 7–9 for risk of malnutrition, and 6 or below for malnutrition [16]. Significant associations were observed between MNA_SF items and AD8 categories (all *p*-values < 0.001) except for psychological stress or acute disease in the past three months, which was evenly distributed between the two AD8 groups (4.0% in cognitive impairment vs. 2.8% in normal cognition; *p* = 0.301). Participants with cognitive impairment reported a higher prevalence of declined food intake compared to those with normal cognition (97.6% vs. 64.6%, respectively). Similarly, a smaller proportion of the cognitive impairment group reported no weight loss compared to the normal cognition group (3.3% vs. 40.2%). Regarding mobility, while 97.2% of participants with cognitive impairment reported going out, only 47.8% of those with normal cognition did so. Moreover, a larger percentage of normal cognition participants had a BMI of 23 kg/m^2^ or greater (89.6%) compared to those with cognitive impairment (47.0%).

No significant associations were found between depression and BMI (*p* = 0.243) or psychological stress (*p* = 0.864). However, participants without depression were more likely to report no decrease in food intake (69.8% vs. 11.8%) and no weight loss (62.8% vs. 13.7%) compared to those with depression. Additionally, all participants in the non-depressed group reported going out (100%) compared to 79.6% in the depressed group.

### 3.5. Multiple Logistic Regression and Potential Predictors of Malnutrition

The results of the simple and multiple regression analyses to identify the *p* predictors of malnutrition are summarized in Table 5. After adjusting for potential confounders, older age (aOR: 1.11, 95% CI: 1.06–1.15), cognitive impairment (aOR: 22.37, 95% CI: 11.32–44.2), and depression (aOR: 12.69, 95% CI: 4.26–37.9) were associated with higher odds of being at risk of malnutrition or being malnourished. Having a caregiver was protective; individuals with caregivers were 33 times less likely to be at risk of malnutrition or malnourished compared to older individuals without caregivers (aOR: 0.03, 95% CI: 0.01–0.1).

## 4. Discussion

Our study is to our knowledge the most recent in the region to report the prevalence of malnutrition and mental health issues in older adults, based on a large-scale sample. Our study showed that 89.5% of community-dwelling older adults at risk of social exclusion and dependency were either malnourished or at risk of malnutrition, 82% were at risk for cognitive impairment and dementia, and 46% had moderate to severe depression in addition to 48% who had mild depression. The study findings also showed that being at risk of dementia and having depression were both significant predictors of malnutrition, independent of other potential confounders.

### 4.1. Prevalence of Malnutrition

In our population of 1410 older adults at risk of dependency, approximately 87% were found to be at risk of malnutrition and 2.5% were classified as malnourished. The prevalence of malnutrition in our study sample is comparable to that reported by other studies on older adults living in the community, whereby recent meta-analyses have estimated it at 3.1% using the same screening tool MNA [29,30]. However, the risk of malnutrition in our study sample is significantly higher compared to the 26.5% reported in a meta-analysis by Cereda et al. [29]. The higher rate of individuals at risk of malnutrition in our study is expected, given that our older adult population, according to the inclusion criteria of poverty and social exclusion, was of lower socioeconomic status —a possible risk factor [31,32,33]. A similarly high rate of 65% was observed in a smaller sample size of 425 older adults of lower socioeconomic status in a study conducted in Bangladesh [32,33].

The socioeconomic status of older adults was worsened by Lebanon’s severe financial crisis at the time of data collection. This crisis, marked by a 36.5% drop in GDP per capita and the country’s reclassification from upper-middle-income to lower-middle-income by the World Bank, directly impacted employment and household incomes [34]. A previous prevalence study conducted in 2014 on community-dwelling older adults living in rural areas, many of whom were of low financial status, reported a significantly lower rate of 37.1% at risk of malnutrition or malnourished, also using the MNA [35]. A slightly higher rate of 61.3% at risk of malnutrition or malnourished was reported in a smaller sample of 111 older adults living in long-term care institutions in 2014 in Lebanon [36]. This suggests a worsening of the nutritional status of older adults over the past ten years in Lebanon.

### 4.2. Prevalence of Cognitive Impairment and Depression

As for mental health, high prevalence rates of both cognitive impairment and depression were also reported in our population. Nearly 80% of the older adults in our study were found to be at risk for cognitive impairment and dementia, a rate significantly higher than the global prevalence of 5.1% to 41% reported in a recent meta-analysis [37]. Among individuals identified with cognitive impairment, 96.7% were either at risk of malnutrition or malnourished as compared to 57.8% in those with normal cognitive function. Other studies on older adults either in institutions or in the community, though with a higher mean age of 75–80 years, have also shown high levels of malnutrition among patients with cognitive impairment, but to a lesser extent [15,38]. The nearly 90% prevalence of malnutrition or risk of malnutrition that we have reported was only reported by a smaller-scale study among older adults with dementia living in institutions rather than in the community, as per a recent meta-analysis [39].

Regarding depression, 94.1% of the subgroup with available data reported experiencing some form of depression, with nearly 45% diagnosed with moderate to severe depression. Similar rates of depression were observed in other countries such as Iran and Greece [40,41,42]. A higher depression rate of 60% was reported in a smaller sample of 221 older adults in Lebanon, although these individuals were institutionalized rather than living in the community [43]. Among individuals diagnosed with depression, 85.7% were either at risk of malnutrition or malnourished, compared to 23.2% of those with normal cognitive function. Very few studies have assessed both malnutrition and depression in specifically older adults. In a recent study conducted in a rural area in Bangladesh, a significantly higher rate of malnutrition was observed when comparing 300 depressed to 300 non-depressed older adults (56% versus 18%, respectively), although this rate is lower than what we reported [44]. Other studies were not conducted amongst older adults specifically.

### 4.3. Exploring the Association of Malnutrition with Mental Health and Potential Predictors

The significantly higher rates of risk of malnutrition in older adults with mental health issues, mainly cognitive impairment and depression, have been further demonstrated in our regression analysis. The prediction model showed that being at risk of dementia and having depression were both significant predictors of either malnutrition or risk of malnutrition, independent of gender and comorbidities, with odds ratios of 22.37 and 12.69, respectively. These odds ratios are substantially higher than that of age, which is 1.11, highlighting the strong impact of dementia and depression on the likelihood of malnutrition. These significant predictors were reported by previous studies such as an association between depression and malnutrition in older adults living in long-term care institutions, as well as a high prevalence of malnutrition among older adults with advanced stages of dementia, both in study populations different from ours, either institutionalized or hospitalized [11,13,36,45]. A recent meta-analysis, which includes studies on older adults living in both nursing homes and the community, with most using the MNA and GDS scales as in our study, also demonstrated a significant association between depression and malnutrition in older adults [46].

The relationship between mental health issues and the deterioration of nutritional status can be attributed to low appetite in depression and swallowing difficulties in dementia, both of which directly impact food intake [30,47]. In fact, the most frequently reported criterion of the MNA SF in our population was decreased food intake since 91.7% reported a moderate or severe decrease. Our study also extended the analysis to examine the association of each MNA_SF criterion with both the risk of dementia and depression. Our analysis highlights significant associations between the MNA_SF “decreased food intake” criterion with both cognitive impairment and depression in older adults. For cognitive impairment, 97% of those with moderate decreases in food intake showed impairment, compared to 61.8% with normal cognition. Similarly, for depression, 86.3% of individuals with moderate decreases in food intake were depressed, compared to 30.2% without depression. Weight loss could not be thoroughly investigated as the majority reported not knowing whether they have lost weight, which suggests that nutritional assessment is not a common practice in older adults and need to be addressed during medical visits. This highlights the necessity for routine nutritional screenings in healthcare settings to ensure the well-being of this vulnerable population [48].

Another complexity of nutritional anthropometrics identified in our study is BMI. While BMI was not significantly associated with depression, higher BMI was significantly reported in those with cognitive impairment, emphasizing the importance of not relying solely on BMI and weight as indicators of nutritional status. This issue remains arguable among clinicians and the non-significant association between BMI and malnutrition is primarily observed in hospitalized patients [49,50]. Older adults may present with increased adiposity and higher BMI despite experiencing muscle loss and malnutrition [51,52]. Thus, our study highlights the importance of thoroughly assessing food intake in older adults with depression and cognitive impairment when screening for malnutrition even in the community setting, rather than relying solely on weight and BMI measurements.

Interestingly, our study found that the absence of a caregiver was associated with a higher rate of malnutrition or risk of malnutrition and that it was a significant independent predictor of risk of malnutrition in the study sample. This can be attributed to the inability of older adults with cognitive impairment and/or depression to independently purchase and prepare food, thus affecting their food intake [53,54]. The role of the caregiver is not well studied when assessing malnutrition and should be a factor to be considered. In the context of our country and other countries of the Middle East region, this factor is of major importance since family support structures are prevalent and older adults usually live with their families and emphasizing this aspect can play a crucial role in addressing malnutrition among the older adult population.

### 4.4. Implications and Recommendations

The high prevalence of risk of malnutrition among older adults with cognitive impairment and depression living at home, along with the significant associations found in our study, raises the need for policies that extend beyond long-term care institutions and hospital settings to support these individuals. This is of particular importance in countries of the Middle East region where a high proportion of older adults stay in their homes rather than move to long-term care institutions due to social norms and cultural practices. Screening for cognitive impairment and depression are usually undertaken as part of health assessments in clinics or primary health care settings; however, nutritional screening is often overlooked [55,56]. It is recommended that these screenings be accompanied by a nutritional status evaluation, or at least an assessment of oral food intake, to enable the earlier detection of malnutrition in this population. Additionally, public health policies should recommend the systematic screening of older adults living in the community, adopting a multi-dimensional approach that includes both mental health and nutrition assessments [30]. This comprehensive approach ensures that malnutrition is identified and addressed promptly alongside mental health issues.

### 4.5. Strengths and Limitations

Our study has several strengths, including a relatively large sample size (n = 1410) and a focus on older adults living with family in the community, unlike many previous studies that concentrated on institutionalized elderly populations. This approach provides valuable insights into groups not receiving continuous professional care and monitoring, which is crucial given the different social and other characteristics between institutionalized and community-dwelling older adults. Furthermore, this study provides the most recent estimates on malnutrition and mental health amongst older adults in Lebanon, whereas previous studies date back at least ten years and were conducted before the recent financial crisis in the country. However, there are some limitations to consider. The cross-sectional and correlational design of this study does not allow for establishing cause-and-effect relationships. Additionally, only a section of the larger sample, specifically those who underwent nutritional assessment, was considered for our study. Due to the nature of the project, this study focused solely on older adults considered vulnerable, either at risk of social exclusion or dependency. Participants were recruited through convenience sampling from primary care facilities, religious institutions, and referrals, which may limit the representativeness of the sample. This approach could exclude older adults who did not meet these criteria or who did not seek institutional support. Furthermore, this study was conducted in the urban setting of Beirut, making the findings less applicable to older adults in more rural areas of the country.

## 5. Conclusions

The present study showed alarming rates of risk of malnutrition, cognitive impairment, and depression among vulnerable community-dwelling older adults in Lebanon mainly at risk of social exclusion and dependency. It also demonstrates the association between criteria of malnutrition, mainly decreased food intake, and these mental health issues and identifies both depression and cognitive impairment as potential predictors of malnutrition in older adults. The strong associations between these conditions underscore the urgent need for multi-dimensional public health strategies that support the integration of nutritional screening into routine health evaluations for older adults, particularly those with cognitive and mental health issues. In regions like the Middle East, where cultural norms favor keeping older adults living within their homes, there is a critical need to implement these policies at the community level.

## Figures and Tables

**Figure 1 geriatrics-09-00122-f001:**
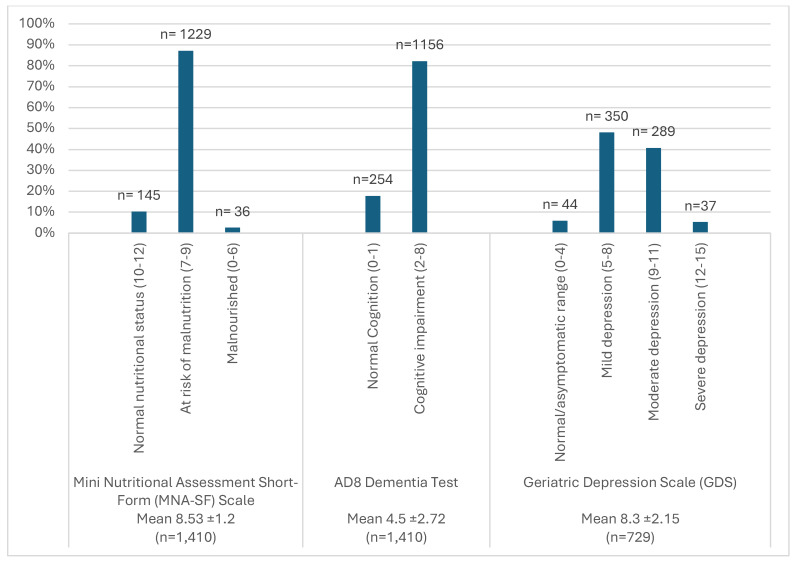
Prevalence rates of malnutrition, cognitive impairment, and depression. MNA_SF: Mini Nutritional Assessment _ Short Form; GDS: Geriatric Depression Score; AD8: Ascertain Dementia score.

**Figure 2 geriatrics-09-00122-f002:**
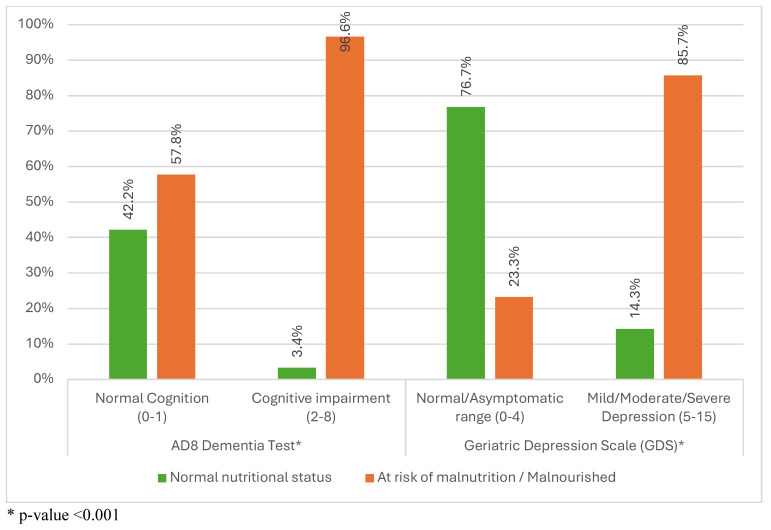
Rates of malnutrition among older adults with cognitive impairment and depression. * *p*-value <0.001; MNA_SF: Mini Nutritional Assessment _ Short Form; GDS: Geriatric Depression Score; AD8: Ascertain Dementia score.

**Table 1 geriatrics-09-00122-t001:** Baseline characteristics of the study population (*n* = 1410).

	*n*	%
Age (years)		
Mean ± SD	71.6	±7.9
Sex		
Female	902	64
Male	508	36
Caregiver		
No	1317	93.4
Yes	93	6.6
Charlson Comorbidity Index		
Cerebrovascular disease	30	2.1
Diabetes	213	15.1
Chronic obstructive pulmonary disease	28	2
Heart failure/chronic heart disease	57	4.1
Dementia	15	1.1
Peripheral vascular disease	27	1.9
Chronic kidney failure (dialysis)	1	0.1
Tumor without metastasis	1	0.1
Charlson Comorbidity Index Score		
0	1158	83.9
1	165	11.9
>1	58	4.3
Recruitment location		
Primary care	723	51.2
Association; NGO	588	41.7
Religious entity	53	3.76
Referral	19	1.35
Other	27	1.91

**Table 2 geriatrics-09-00122-t002:** Prevalence of items of the Mini Nutritional Assessment _ Short Form (MNA_SF) scale (*n* = 1410).

	*n*	%
Has food intake declined over the past 3 months due to loss of appetite, digestive problems, chewing, or swallowing difficulties?		
Severe decrease in food intake	14	1.0
Moderate decrease in food intake	1279	90.7
No decrease in food intake	117	8.3
Recent weight loss (<3 months)		
Weight loss > 3 kg	6	0.4
Does not know	1247	88.4
Weight loss between 1 and 3 kg	18	1.3
No weight loss	139	9.9
Mobility		
Bed or chair bound	22	1.6
Able to get out of bed/chair but not able to go out	138	9.8
Able to go out	1250	88.7
Has suffered psychological stress or acute disease in the past 3 months		
Yes	42	3.0
No	1368	97.0
Body mass index (BMI = weight/(height) 2 in kg/m^2^)		
Mean ± SD	24.2	±3.7
0 = BMI less than 19	9	0.6
1 = BMI 19 to less than 21	128	9.1
2 = BMI 21 to less than 23	503	35.7
3 = BMI 23 or greater	770	54.6
MNA total score		
Mean ± SD	8.53	
≥12: normal nutrition status	145	10.3
8–11: at nutritional risk	1229	87.2
≤7: malnourished	36	2.5

**Table 3 geriatrics-09-00122-t003:** Association of the modified Mini Nutritional Assessment _Short Form (MNA_SF) criteria with Ascertain Dementia (AD8) Score.

	Total n	AD8				*p*-Value
		Normal Cognition (0–1)		Cognitive Impairment (2–8)		
		*n*	%	*n*	%	
Mini Nutritional Assessment_Short Form (MNA_SF) total score						
Normal nutritional status (10–12)	145	106	42.2	39	3.4	<0.001
At risk of malnutrition (7–9)	1229	132	52.6	1097	94.7	
Malnourished (0–6)	36	13	5.2	23	2	
MNA_SF criteria						
Has food intake declined over the past 3 months due to loss of appetite?						
Severe decrease in food intake	14	7	2.8	7	0.6	<0.001
Moderate decrease in food intake	1279	155	61.8	1124	97.0	
No decrease in food intake	117	89	35.5	28	2.4	
Recent weight loss (<3 months)						
Weight loss > 3 kg	6	1	0.4	5	0.4	<0.001
Does not know	1247	140	55.8	1107	95.5	
Weight loss between 1 and 3 kg	18	9	3.6	9	0.8	
No weight loss	139	101	40.2	38	3.3	
Mobility						
Bed or chair bound	22	18	7.2	4	0.3	<0.001
Able to get out of bed/chair but not able to go out	138	113	45.0	25	2.2	
Able to go out	1250	120	47.8	1130	97.5	
Has suffered psychological stress or acute disease in the past 3 months						
Yes	42	10	4.0	32	2.8	0.301
No	1368	241	96.0	1127	97.2	
BMI						
0 = BMI less than 19	9	3	1.2	6	0.5	<0.001
1 = BMI 19 to less than 21	128	8	3.2	120	10.4	
2 = BMI 21 to less than 23	503	15	6.0	488	42.1	
3 = BMI 23 or greater	770	225	89.6	545	47.0	

BMI: Body Mass Index. AD8: Ascertain Dementia score; MNA_SF: Mini Nutritional Assessment _ Short Form.

**Table 4 geriatrics-09-00122-t004:** Association of the modified Mini Nutritional Assessment_Short Form MNA_ SF Criteria with Geriatric Depression Score GDS.

	Total n	GDS				*p*-Value
		Normal (0–4)		Mild/ Moderate/Severe Depression (5–15)		
		*n*	%	*n*	%	
Mini Nutritional Assessment_Short Form (MN_-SF) total score						
Normal nutritional status (10–12)	131	33	76.7	98	14.3	<0.001
At risk of malnutrition (7–9)	570	9	20.9	561	81.8	
Malnourished (0–6)	28	1	2.3	27	3.9	
MNA_SF criteria						
Has food intake declined over the past 3 months due to loss appetite						
Severe decrease in food intake	13	0	0.0	13	1.9	<0.001
Moderate decrease in food intake	605	13	30.2	592	86.3	
No decrease in food intake	111	30	69.8	81	11.8	
Recent weight loss (<3 months)						
Weight loss > 3 kg	5	0	0.0	5	0.7	<0.001
Does not know	587	13	30.2	574	83.7	
Weight loss between 1 and 3 kg	16	3	7.0	13	1.9	
No weight loss	121	27	62.8	94	13.7	
Mobility						
Bed or chair bound	19	0	0.0	19	2.8	0.004
Able to go out of bed/chair but not goes out	121	0	0.0	121	17.6	
Goes out	589	43	100.0	546	79.6	
Has suffered psychological stress or acute disease in the past 3 months						
Yes	38	2	4.7	36	5.2	0.864
No	691	41	95.3	650	94.8	
BMI						
0 = BMI less than 19	2	0	0.0	2	0.3	0.363
1 = BMI 19 to less than 21	56	2	4.7	54	7.9	
2 = BMI 21 to less than 23	203	8	18.6	195	28.4	
3 = BMI 23 or greater	468	33	76.7	435	63.4	

BMI: Body Mass Index; GDS: Geriatric Depression Score; MNA_SF: Mini Nutritional Assessment _ Short Form.

**Table 5 geriatrics-09-00122-t005:** Predictors for malnutrition based on simple and multiple regression analyses.

	*n*	Nutritional Status Based on MNA_SF (At Risk of Malnutrition/Malnourished vs. Normal Nutritional Status)							
		Simple				Multiple			
		OR	*p*-Value	95%CI		aOR	*p*-Value	95%CI	
Age									
Mean ± SD	1410	1.09	<0.001	1.06	1.12	1.11	<0.001	1.06	1.15
Sex									
Female	902								
Male	508	0.94	0.748	0.66	1.4	0.66	0.144	0.38	1.2
Caregiver									
No	1317								
Yes	93	0.1	<0.001	0.06	0.2	0.03	<0.001	0.01	0.1
Charlson Comorbidity Index									
0	1158								
1	165	1	0.964	1	1.7	2	0.380	1	4.0
2+	58	0.41	0.008	0.21	0.8	0.98	0.968	0.34	2.8
AD8 Dementia Test									
Normal cognition (0–1)	251								
Cognitive impairment (2–8)	1159	20.99	<0.001	13.99	31.5	22.37	<0.001	11.32	44.2
Geriatric Depression Scale (GDS)									
Normal (0–4)	43								
Mild/moderate/severe depression (5–15)	686	19.8	<0.001	9.46	41.5	12.69	<0.001	4.26	37.9

MNA_SF: Mini Nutritional Assessment _ Short Form; GDS: Geriatric Depression Score; AD8: Ascertain Dementia score. aOR: adjusted Odd Ratio.

## Data Availability

The original contributions presented in the study are included in the article; further inquiries can be directed to the corresponding author.

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
