# Peer review of "The Association of Cognitive Impairment and Depression with Malnutrition among Vulnerable, Community-Dwelling Older Adults: A Large Cross-Sectional Study"

_geriatrics, 2024, doi:10.3390/geriatrics9050122_

Round 1

Reviewer 1 Report

Comments and Suggestions for Authors

The authors conducted a cross-sectional study to evaluate the association of cognitive impairment and depression with malnutrition among community-dwelling older adults who were dependent or at risk of dependency as determined by the INICIARE tool and/or at risk of poverty and social exclusion as determined by the AROPE tool. The authors performed secondary data analyses of a subset of the TEC-MED project cohort, involving 1,410 older adults aged 60 and above in Beirut. They report that in their regression model, older age, cognitive impairment, and depression were significant predictors of being malnourished or at risk of malnutrition, and having a caregiver was a protective factor. The topic is of great public health importance and the manuscript is overall well written. Suggestions:

1) The title suggests that the focus is on "community-dwelling older adults"; however, the study has intentionally selected vulnerable older adults who are dependent and/or at risk of poverty & social exclusion. Therefore, findings do not apply to the general older adult population in the community. Please clarify the target population in the title, in the abstract, and throughout the manuscript.

2) Attrition is alarmingly high- out of 4259 participants, data is only available in 1410 participants. Please explain reasons for such a high attrition rate. If possible, please consider doing a sensitivity analyses such as inverse probability weighting to address selection bias from attrition. If sensitivity analyses are not possible, please at least discuss this as a limitation.

3) The outcome is a composite of "at risk of malnutrition" + "malnourished"; yet in the manuscript and in the abstract, the authors have labelled this composite outcome as "malnourished" which is inaccurate. As stated in line 198 in section 3.2, malnutrition was found in only 2.5% individuals. Please clarify this in the abstract and rewrite the conclusion of the abstract and the conclusion in the main manuscript to accurately reflect the composite and not "malnourished". Please also make this change throughout the manuscript as appropriate.

4) The TEC-MED project is across several countries- if other countries are included, you may have better statistical power to look at malnutrition, instead of looking at a composite outcome of "at risk of malnutrition" + "malnourished".

5) Considering that the cohort included special, vulnerable older adults, the prevalence estimates may not apply to the general older adult population in the community. Therefore, please specify in the results and discussion that the prevalence estimates only apply to vulnerable older adults.

6) With regards to depression and malnutrition, no systematic reviews/meta-analyses have been cited and discussed. Please consider adding it to the discussion. For example: Hu W, Mao H, Guan S, Jin J, Xu D. Systematic Review and Meta-analysis of the Association Between Malnutrition and Risk of Depression in the Elderly. Alpha Psychiatry. 2024 Mar 1;25(2):183-189. doi: 10.5152/alphapsychiatry.2024.231336. PMID: 38798803; PMCID: PMC11117414. 

7) In the limitations para, please mention limitations such as selection bias (response bias & attrition), recall bias and potential for misclassification of exposure and outcomes, and unmeasured and residual confounding. 

Thank you for your submission and contributions to an important topic.

Author Response

Dear reviewer,

Thank you for your thorough review, insightful comments and recognizing the relevance of our article's focus on aging and nutrition as a crucial public health area. We have revised the article based on the new comments, addressing each one in detail and highlighting the changes done in the paragraphs below. The changes made in response to the feedback are also in the updated manuscript.

Reply to comments

Comment 1: The comment is of great importance and therefore the title was adjusted to The Association of Cognitive Impairment and Depression with Malnutrition among Vulnerable, Community-Dwelling Older Adults : A Large Cross-Sectional Study

In addition, the vulnerability was highlighted in the abstract mainly in background and conclusion part. In the discussion, it was mentioned in the first paragraph (line 305) to rehighlight it and discussed again in the limitation (line 452). It was emphasized again in the conclusion (line 461).

Comment 2: The whole TECMED project was on 4259 participants but nutritional assessment was only done on 1410 participants who accepted to do more thorough assessment. The scope of our study is only to explore the association of malnutrition with mental health in these older adults. Therefore the sample size for our study sepcifcally was only 1410. We have highlighted this sample size in our revision of the manuscript under study design (line 112). Sinc eit is not considered an drop out sensitivity analysis could not have been done. Therefore the point was added to the limitations as suggested (line 450)

Comment 3: This is a crucial comment. We have presented the results as at risk and malnourished both in mansucript and abstract. Changes have been done accordingly in the abstract conclusion and in the conclusion of the manuscript (line 46) to avoid this confusion. Other mentions of the risk versus malnutrition in the manuscript were done in the first paragraph of the discussion (line 305), it was already highlighted in the second paragraoh of the discussion (line 320 ). The risk of malnutrition was clarified again and amneded in the part of implications (line 423)

Comment 4: Based on TEC-MED project guidelines, each country was responsible of having ethical approval and following on publication of the data collected. There was no collaboration in combining statiscal analysis for malnutrition and other criteria.

Comment 5: The vulnerapility of the sample was highlighted in the sample size as inclusion and exclusion criteria. In repsone to comment 1, chnages have already been made to highligh the point on vulnerability in the first part of discussion, the limiation and the conclusion. Lines mentioned in comment 1.

Comment 6: The reference cited if of great relevance and(Hu et al., 2024) it was added to the discussion in parapfarh 6 (line 346).

Comment 7: The whole section on limitation was amended to include the previous comments related to sample size, vulnerability and the points mentioned here related to exposure and outcomes. (lines 452-458)

Thank you once again for your detailed insights, which were very helpful in revising the manuscript, particularly in clarifying key points in the methodology and addressing the limitations.

Reviewer 2 Report

Comments and Suggestions for Authors

Dear Author(s),

Thank you for the opportunity to review the article titled "The Association of Cognitive Impairment and Depression with Malnutrition among Community-Dwelling Older Adults: A Large Cross-Sectional Study."

Here are my comments on the manuscript:

Global trends indicate a rapid and significant aging of the population. Aging is a natural process in every person's life, bringing numerous changes, with mental status changes and nutritional disorders playing a leading role. These depend on various factors—socioeconomic, geographical, cultural, etc.—and understanding them allows for the timely development of appropriate treatment and prevention strategies.

Therefore, the topic of this article is relevant, which is reflected in the title of the text, clearly and precisely defining the research objective.

General Comment on the Article
I suggest a language review of the text and the correction of grammatical errors such as excessive spacing between letters (for example, in "global prevalence," line 40), or correcting typos (e.g., "conginitve" in lines 45, 60, and "arranegements" in line 72), as well as stylistic and grammatical improvements.

The article's abstract outlines the key features of the research and is written in accordance with the guidelines.

1. Introduction
I suggest that the authors comment on the data mentioned in lines 53-55: "Based on available studies...". Specifically, there is an extremely wide range of dementia prevalence from 6% to 75%. What causes such a disparity in the results of different studies? The issue likely lies in methodology, socioeconomic, and geographical differences...

Please rephrase the sentences in lines 55-57: "Studies...." and "Depression..." and change their order. Simplify and rephrase the sentences in lines 69-74. It concerns how the cultural characteristics of Lebanon lead to older people staying in a home environment and rarely being placed in nursing homes.

In line 83, the word "Dementia" appears without any context or inclusion in a sentence.

The study's goal (lines 80-82) is to determine the basic epidemiological characteristics of dementia and malnutrition prevalence and their interrelationship to establish guidelines for their early detection and timely treatment. The sentence provided by the authors needs refinement.

2. Materials and Methods
2.1 Study Design and Population
I suggest revising and simplifying the entire section—lines 86-111.
For example, split the sentence in lines 86-90 into two sentences ("This observational…"). This will clarify your intent and make it easier to read.

In lines 90-93, the acronym "TEC-MED" is mentioned twice in the sentence. I suggest simplifying this sentence as well.

Indicate which version of the INICIARE questionnaire was used in the study, long or short form—I assume the short form.

In the introduction, the authors mention that the TEC-MED study included 4,259 participants. Please state here how many participants were included specifically in this study based on inclusion and exclusion criteria (the authors mention this in the results, line 173, section sample description).

2.2 Data Collection
The previous section discusses how the TEC-MED study was part of an international project from 2014-2020. Here, it states that data were collected between 2022 and 2023.
I suggest the authors clarify, and simplify sections 2.1 and 2.2.

Sections 2.3 and 2.4 describe the use of questionnaires.
The MNA SF was used, and one question was omitted with an explanation, and accordingly, the scoring system was adjusted, which can be discussed but is understandable from the authors' perspective.

The GDS 15 is a standard questionnaire. However, if the questionnaire scoring is followed, a discrepancy arises. According to https://www.physio-pedia.com/Geriatric_Depression_Scale, the scoring system used in this article is obtained. However, according to The Hartford Institute for Geriatric Nursing, New York University, College of Nursing, the scoring system implies suspicion of depression with a score of >5, and clear depression at ≥10 (https://www.academia.edu/86609929/The_Geriatric_Depression_Scale_GDS_) . Scores of 6-10 indicate mild to moderate depression, while scores of 11-15 indicate severe depression (Friedman et al., 2005).

It would be good to cite the source on which this study is based, as different results could be obtained.

3.2. Prevalence of Malnutrition
The authors state in lines 188 and 189: "Table 2 presents the Mini Nutritional Assessment Short Form (MNA-SF) scale items and its total score." However, the total score is not mentioned in that table; instead, the number of participants according to their responses to individual questions is listed. Generally, this table is redundant, as the overall score achieved in the MNA SF questionnaire is relevant. If the authors wish to analyze the individual responses in the questionnaire, they can mention this textually.
Additionally, the authors have modified this questionnaire, so in this sentence and later mentions, including the table title, it should be referred to as the "modified MNA SF questionnaire."

3.3. Prevalence of Cognitive Impairment and Depression
Although the section title clearly defines that it deals with the prevalence of cognitive impairment and depression, malnutrition appears in Figure 1 under the title "Prevalence rates of Malnutrition, Cognitive Impairment, and Depression," which is out of place here. Either change the section title and combine 3.2 and 3.3, or separate the first part of Figure 1 dealing with the MNA score into a separate figure, which would be better.

Also, I reiterate that it must be stated that this is a "modified MNA SF score," as any reader familiar with the questionnaire will be confused when viewing the categories in Figure 1, which lists scores of "malnourished 0-6, at risk of malnutrition 7-9, normal nutritional status 10-12." These scores have been modified by the authors, and I suggest removing the numbers in the categories from the figure.

The authors might consider including the absolute values (N) in the columns. Namely, percentages are already listed on the X-axis.

3.4. Rates of Malnutrition among Older Adults with Cognitive Impairment and Depression
It's not quite appropriate to use the phrase "Regarding depression, among those in the normal or asymptomatic range" (line 231). Asymptomatic can also include patients with depression. It concerns participants with a normal score, without "asymptomatic."

In Figure 2, the number 96.6% has been shifted during preparation, but I assume it's due to "shortening the graphical presentation."

3.5. Association of Malnutrition and MNA Criteria with Cognitive Impairment and Depression
Again, I emphasize the term MNA SF and "modified questionnaire." Please correct the language errors.

4. Discussion
I suggest the authors divide the discussion into several chapters—malnutrition, cognitive impairment, depression, and then the relationship between them, subdivided into subheadings. This will make the discussion more organized and readable.

Below are just a few minor corrections and suggestions:

  • Line 301: "80% were at risk for cognitive impairment and dementia" – 82% have cognitive impairment.
  • Line 301,302: "45% had moderate to severe depression" – 46% have it, and additionally, according to the criteria used, another 48% have mild depression. This totals 94%, which is an extremely high number and certainly requires comments from the authors.
  • Line 310: "using the same screening tool MNA" – it is true that those studies used the questionnaire, but they did not modify it as the authors of this text did. It would be beneficial for the authors to address this fact
  • Line 322: "...incomes (World..." This seems to be only part of an intended literature reference — needs correction.
  • Line 328: "This suggests a worsening of the nutritional status of older adults over the past ten years in Lebanon" – the authors obtained a figure of 90% of participants (the text says 87+2.5%, and in Figure 87 + 3% - needs alignment) at risk of malnutrition or malnourished. This is an extremely high number of participants affected by malnutrition or at risk.

Author Response

Dear reviewer,

Thank you for your thorough review, insightful comments and recognizing the relevance of our article's focus on aging, mental health, and nutrition. We have revised the article based on the new comments, addressing each one in detail and highlighting the changes done in the paragraphs below. The changes made in response to the feedback are also in the updated manuscript.

Reply to "General comment on the article":

We have read again the article and did all necessary corrections of format and grammar,  the ones mentioned and other ones after proof reading (such as indicated and in line 143). Excessive spacing was also removed (such in line 78 and 82).

Reply to Introduction:

Comment 1: It is very important to explain discrepancies of prevalence rates. A sentence was added accordingly with a relevant reference. Lines 56-58: "This wide discrepancy in prevalence can be attributed to the use of different diagnostic tools for assessing malnutrition, as well as variations in socioeconomic conditions and geographic locations across study populations"

Comment 2: Sentences were rephrased to better protray the culture context. Lines 72-78 "This aging process is supported by strong social norms that encourage families to do living arrangements to keep their older adults at home priorotize cultural habits of having older adults living with the family or in their own homes (Hussein & Ismail, 2017; Shideed et al., 2013). These efforts lead to arrangements like multi-generational households, family caregivers, or home-based nurses, contributing to a higher proportion of older adults living in the community rather than in long-term care facilities (Shideed et al., 2013)."

Comment 3: We apologize for the mistake and the word Dementia was removed.

Comment 4: The 2nd goal was rephrased to have an explicit objective as suggested. Lines 81-83 "determining the basic epidemiological indicators of dementia, depression, and malnutrition, and investigating their association to guide early detection and timely treatment".

Reply to Materials and Methods:

Study Design and Population:

Comments 1, 2, 3 and 4: The first two paragraphs were revised and simplified to have a clearer content. It was also mentioned that the short form of INCIARE was used and the final number of participants included in our study specifically based on participants who had their nutritional assessment.  Lines 89-113

" This observational, community-based cross-sectional study uses secondary data from Lebanon’s cohort within the Transcultural Socio-Ethical-Care Model for Dependent Populations in the Mediterranean Basin (TEC-MED), which is part of the 2019-2023 CrossBorder Cooperation (CBC) initiative, funded by the European Neighbourhood Instrument (ENI) (Gálvez A, 2024). The project aims to create a transcultural care approach for dependent elderly people in six Mediterranean countries (Spain, Greece, Italy, Tunisia, Egypt, and Lebanon) and includes an initial assessment using an integrated platform, followed by social care interventions.

In Lebanon, participants were recruited through convenience sampling from primary care centers, religious institutions, and referrals in the capital Beirut. Eligible participants were older adults (60 years and above) living at home and classified as dependent or at risk of dependency, based Inventario del NIvel de Cuidados mediante Indicadores de Resultados de Enfermería (INICIARE) and/or at risk of poverty and social exclusion by At Risk of Poverty or Social Exclusion (AROPE) criteria. The INICIARE tool short from was used to measure dependency levels by assessing functional capabilities and support needs, including feeding, continence, and health behavior (Morales-Asencio et al., 2015). AROPE is an EU policy tool used to assess individuals or households at risk of poverty, severe material deprivation, or living in households with low socioeconomic status (Gonzalez et al., 2021). Exclusion criteria included severe or total dependency according to INICIARE and/or AROPE, needing palliative or end-of-life care, uncorrected severe hearing loss, uncontrolled severe mental illness and those taking major tranquilizers or other drugs affecting cognitive status. A total of 4,259 participants met the criteria and were enrolled in the TEC-MED project in Lebanon. However, our study specifically included only the 1,410 participants who completed the nutritional assessment."

Data collection:

Comment 1: The TEC-MED project, part of the CBC initiative, ran from 2019 to 2023 (link to project: https://www.enicbcmed.eu/projects/tec-med). Since the CBC initiative began in 2014, the study design was updated to reflect only the TEC-MED project dates to avoid any confusion regarding timelines (lines 90-93:) The section was also reformulated for clearer description. Lines  116-122

"Data collection occurred between April 2022 and February 2023 through home visits by trained social workers, who conducted an initial assessment. The social workers underwent comprehensive training by physicians and dietitians on conducting nutritional assessment and using the to assess mental health, with a focus on cognitive impairment and depression. When applicable, caregivers providing care to these older persons at home were also involved in the data collection. Data on the patient’s characteristics, including age, gender, comorbidities and presence of caregiver were also recorded (Morales-Asencio et al., 2015)."

Comment 2: The question was omitted, and the scale was modified only when analyzing associations and conducting correlation and regression tests. However, the full scale was used to measure and report the prevalence of malnutrition. This modified score in the regression analysis was applied to prevent confounding with dementia and depression, as the omitted MNA question specifically addresses these conditions. This approach has been used in previous studies examining similar outcomes, notably Sanders et al. (2016). 
The sentence was reformulated to clarify that the question was omitted only during regression tests and it was added as a section under statistical analysis to avoid confusion. Line 158-165

"When analyzing the association between malnutrition and dementia, the scale was modified by removing the dementia and depression item (scored 0-2). Based on Sanders et al., the score cutoffs were adjusted: 10-12 for normal nutrition, 7-9 for risk of malnutrition, and 6 or below for malnutrition (Sanders et al., 2016). This modification was applied only for correlation and regression models to avoid confounding elements."

Comment 3: GDS was used with the following scale Scores ≤ 4 were considered normal, 5-8 indicative of mild depressive symptoms, 9-11 moderate, and ≥ 12 severe depressive symptoms as first published by Yesavage and then validated in the Lebanese population in arabic by one of our authors Karam et al 2018. Therefore we have used this scale and we have also cited the references in the end of the paragraph as recommended to have a clearer source. Lines 145-147

Reply to Results:

Prevalence of malnutrition:

Comment 1: MNA total score was added to Table 2 in order to reflect the title as recommended. Title has also been adjusted as follows Prevalence of items of the Mini Nutritional Assessment Short-Form (MNA-SF) Scale. The total score is also shown in Figure 1 . We recommend to keep the different items of MNA score also listed in table and not only in text because they will be used as individual criteria in our regression model with dementia and depression to identify predictors. In addition the BMI and food intake items are also part of our discussion. 

Comment 2: As explained in the previous response on methodology, the full MNA score was used to measure prevalence, and all MNA items were collected. The modified version was only applied in the statistical analysis, specifically in the regression model, to prevent confounding factors, following the approach used in previous studies. We have clarified this in our methodology section to avoid any confusion as recommended.

Prevalence of Cognitive Impairment and Depression

Comments 1 and 3: The two sections 3.2 and 3.3 were combined with a title of Prevalence of malnutrition, cognitive impairment and depression in order to have the figure reflection the section as suggested. The N numbers were added in the column as suggested.

Comment 2: As explained in the previous response on methodology and in the above comment, the full MNA score was used to measure prevalence, and all MNA items were collected. The modified version was only applied in the statistical analysis, specifically in the regression model, to prevent confounding factors, following the approach used in previous studies. Therefore the scores were kept the same. We have clarified this in our methodology section to avoid any confusion as recommended.

Rates of Malnutrition among older adults with cognitive impairment and depression

Comment 1: term normal or asymptomatic range was replaced with normal score as recommended.

Comment 2: it is a shifting happening during formatting. 

Association of Malnutrition and MNA criteria with cognitive impairment.

The title was changed to Association of malnutrition with modified  MNA-SF and MNA-SF criteria with cognitive impairment and depression. Titles of tables 3 and 4 were also adjsuted using modified MNA-SF

In order or highlight again that the modified MNA-SF was only used in association and regression analysis, it was added to the paragraph under the section. Lines 252-255 "The modified MNA scale, omitting the dementia and depression item  was used when analyzing associations with dementia, and the score cutoffs were adjusted 10-12 for normal nutrition, 7-9 for risk of malnutrition, and 6 or below for malnutrition .(Sanders et al., 2016)"

Reply to Discussion:

Comment 1: subheadings were added to have more organized and readable discussion as suggested 

All corrections were made as follows:

Comment to Line 301: the 80% was corrected with 82% (Line 306)

Comment to Line 302: the mild depression rate of 48% was added (line 308) as for the total rate of 94% it was already mentioned and discussed thoroughly in the discussion paragraph 5 (line 351)

Comment to Line 310: The MNA-SF with no modification as explained in the above replies was used to measure prevalence of malnutrition and the modified was only used in regression and correlation to avoid confounding elements. Therefore we can mention in the discussion that the same tool of MNA-SF was used.

Comment to Line 322: Reference is World Bank and it was corrected.(line 329)

Comment to Line 328: the 90% was aligned with results and corrected to 89.5% and it was even corrected in the first paragraph of the discussion (line 305)

Thank you once again for your thorough insights, which were incredibly helpful in revising the manuscript accordingly.